# Partners at Care Transitions (PACT) — exploring older peoples' experiences of transitioning from hospital to home in the UK: protocol for an observation and interview study of older people and their families to understand patient experience and involvement in care at transitions

Natasha Kate Hardicre,[1] Yvonne Birks,[2] Jenni Murray,[1] Laura Sheard,[1] Lesley Hughes,[1] Jane Heyhoe,[1] Alison Cracknell,[3] Rebecca Lawton[4]

For numbered affiliations see end of article.

**Correspondence to**
Dr Natasha Kate Hardicre; natasha.hardicre@bthft.nhs.uk

## ABSTRACT

**Introduction** Length of hospital inpatient stays have reduced. This benefits patients, who prefer to be at home, and hospitals, which can treat more people when stays are shorter. Patients may, however, leave hospital sicker, with ongoing care needs. The transition period from hospital to home can be risky, particularly for older patients with complex health and social needs. Improving patient experience, especially through greater patient involvement, may improve outcomes for patients and is a key indicator of care quality and safety. In this research, we aim to: capture the experiences of older patients and their families during the transition from hospital to home, and identify opportunities for greater patient involvement in care, particularly where this contributes to greater individual-level and organisational-level resilience.

**Methods and analysis** A 'focused ethnography' comprising observations, 'Go-Along' and semistructured interviews will be used to capture patient and carer experiences during different points in the care transition from admission to 90 days after discharge. We will recruit 30 patients and their carers from six hospital departments across two National Health Service (NHS) Trusts. Analysis of observations and interviews will use a framework approach to identify themes to understand the experience of transitions and generate ideas about how patients could be more actively involved in their care. This will include exploring what 'good' care at transitions looks like and seeking out examples of success, as well as recommendations for improvement.

**Ethics and dissemination** Ethical approval was received from the NHS Research Ethics Committee in Wales. The research findings will add to a growing body of knowledge about patient experience of transitions, in particular providing insight into the experiences of patients and carers throughout the transitions process, in 'real

## Strengths and limitations of this study

► Using a range of qualitative methods, the study will generate rich, in-depth data to contextualise patient involvement and experiences of transitions of care from hospital admission and throughout the transition period, from the point of view of older people and their carers.
► The longitudinal approach enables us to gain insight into how patient experience and involvement change over time.
► While the study design enables in-depth data to be captured from a small number of older people and their carers, as is the nature of qualitative inquiry, this limits generalisability of findings. The study is situated within a larger programme which will allow greater generalisability, as the programme of work progresses informed by this phase.
► Although non-participant observation can generate rich contextual data that are not as easily accessed via other methods, the presence of a researcher has the potential to affect the behaviour of those being observed.

time'. Importantly, the data will be used to inform the development of a patient-centred intervention to improve the quality and safety of transitions.

## INTRODUCTION

Reduced lengths of stay in hospital can result in patients being discharged from hospital to home with ongoing treatment and care needs. Shorter stays in hospital have benefits for both patients, who prefer to be at

home, and hospitals, which can treat more patients if stays are shorter. However, reduced stays can also result in an increased reliance on care outside the inpatient setting, for example, wound or catheter care, changes to medication or input from therapy services. 'Discharge' from hospital is, therefore, more likely to be a stage in a process involving the *transfer* of care, rather than being an *end point* of care. The movement and transfer of care from hospital to home—sometimes referred to as the 'transition period'—is likely to involve input from multiple agencies to meet patients' ongoing care needs. It is a highly variable and complex process that is contingent on several factors, for example, service provision, resource capacity and knowledge transfer within and between secondary care teams, general practitioners (GPs) and corollary services, community therapy teams and adult social care services,[1] alongside the social support networks and resources that patients themselves have access to (or not). Consequently, the transition of care from hospital into community settings can be a risky one. Additionally, older people may experience more than one 'transition' in a single hospital admission episode, for example, moving between wards or via intermediate care at a different location. Likewise, some older people may experience readmissions within a short period of time. The transitions process may not, therefore, be a linear one, resulting in further complexity.

As many as one in five patients experience an adverse event in the transition from hospital to home, 62% of which could be prevented[2]; this is double the number of adverse events experienced by patients during a hospital stay.[3] For older patients, who are more likely to have complex health and social needs, and who may be anxious, confused and disorientated,[4,5] the risks associated with transitions of care may be greater than that of the general population. This may result in a higher-than-average rate of readmission to hospital,[6] thereby prolonging the overall patient stay. This counteracts the benefit of reduced patient stays, and further exposes patients to risks associated with being in hospital. Krumholtz[7] argues, for example, that hospitalisation causes 'substantial stress' to patients, through causes such as disrupted sleep, poor nourishment, 'a baffling array of mentally challenging situations', changes to medication and deconditioning associated with inactivity and bed rest. Older people are particularly vulnerable to such stressors as they are more likely to have multiple morbidities, take multiple medications and remain inactive.[8,9] Moreover, older people are the highest users of the National Health Service (NHS) and, with the number of people in the UK aged 75 and over set to double in the next 30 years, this group of patients is an important target for support.[10] Increased risk associated with both hospitalisation and the transition period suggests that improving the quality and safety of care during this time is critical.

Patient experience of care is a key indicator of quality and safety[11] and so an important target for intervention. Indeed, this strong relationship between patient experience and outcomes suggests that those interested in improving health outcomes (quality, safety and cost savings) should strive first to improve patient experiences, especially by focusing on activities such as patient engagement. However, despite a growing emphasis on shared care and patient empowerment,[12] the involvement of patients in their care before, during and after transitions remains minimal, with patients feeling that they are not always listened to and that they did not have a 'lot of say' in their care.[13–19] A recent systematic review of patient experiences of transitions highlighted the necessity of involving older people and their carers in the discharge process, but reported variability in the degree to which this was achieved.[20] The study described in this protocol forms the first of six interlinked 'work packages' (WP) in a National Institute for Health Research funded Programme Grant for Applied Health Research that aims to understand and improve the experience, and safety, of care for older patients during transitions and, by doing so, reduce readmissions and NHS costs. In particular, we want to explore whether greater involvement of patients and their families can improve patient experience and safety at the transitions of care. This will involve exploring patient experience of transitions and using these data to develop and test a patient-centred intervention that supports the involvement of older people, and their families, in their care.

There are several published studies that have explored patient and carer perspectives on care at transitions.[13–20] However, much of this work appears to capture people's experiences at a single time point, often retrospectively after discharge. The study outlined here will, instead, recruit people while in hospital, and follow them until approximately 3 months postdischarge. The longitudinal nature of the study will enable us to capture continuity and change in experience and involvement over time and will thus contribute new data and findings to a growing body of literature on care at transitions. Moreover, the programme of work uses a resilience engineering approach to safety in healthcare.[21] We especially want to learn from what goes well at transitions, rather than focus only on what goes wrong; doing so 'sheds light on otherwise unrecognised and unspecified pathways to success'.[22] Within this project, we want to understand resilience at two levels: (1) how patients and carers themselves bounce back, adapt and essentially cope with the transition process and what helps them to do this; and (2) how do patients and relatives get involved to prop up the transition process; in other words, what work do they and their informal and formal carers do to adapt to and overcome obstacles arising from a less-than-ideal system (eg, discharge letters arriving at primary care days after discharge). In this latter case, we will explore the ways that those people involved in the transitions process contribute to system resilience. Schubert *et al*,[23] for example, suggest that patients/caregivers can 'identify and prevent mistakes from happening, and participate in improving their care' by navigating a 'fragmented system'

through the coordination of tasks across multiple healthcare settings and providers. This will enable us to take a proactive approach towards care during the transitions period, developing an intervention that helps to support older people to be more involved in the transition and so make the transitions process 'good'. We believe this is a novel approach towards understanding and improving care at the transitions period.

The research study described here focuses on understanding the transitions process from the perspective of those experiencing it—patients and their families. There are two main foci of the research:
1. Experience: Describing the transition process from the point of view of older patients and their carers;
2. Involvement: Exploring where the opportunities are for improving patient involvement in the transition process.

Research questions are as follows:
1. What do patients and their families experience during the transition of care from hospital to place of residence?
2. What do patients think, feel and believe about this process?
3. How can people be more involved in their care:
   a. To what extent do people feel involved in their care? What are their perspectives on this?
   b. Where are the opportunities for patients to be more involved in their care?
   c. To what extent do people feel able to be (more) involved in their care? What has helped, or would help, them to feel able to be (more) involved in their care?

## METHODS AND ANALYSIS
### Recruiting patients
Beginning in May 2017, 30 older patients (aged 75+), and their immediate carers, will be recruited to the study. Patients and carers will be recruited from six departments specialising in elderly medical care, respiratory care, orthopaedic care of the elderly and stroke, across two hospitals. The departments have been selected for the study to reflect different transitional challenges, emergency and elective admissions (including elective surgery), acute and chronic illness and multimorbidity or polypharmacy issues.

Sampling aims to capture 'maximum variation' in respondents.[24] We will purposively aim to recruit a diverse group of patients from different ethnicities, and gender groups, as well as a variety of ages—including the 'oldest old' (aged 85+)—wherever possible. We will also try to ensure that people with and without carers are included in the research, as carer involvement is likely to have an impact on the patient's experience of transition. Although sampling will be purposive, we recognise that in this context and population, there is likely to be a degree of opportunistic recruitment; initially, the researchers will speak to clinical staff on each ward to identify eligible

patients, selecting those who meet the criteria and who are available to approach at that time. The diversity of the sample will be monitored as participants are recruited. We anticipate that a sample of 30 patients is likely to allow us to capture some diversity and is also likely to achieve theoretical saturation; however, this will be reviewed as analysis proceeds to ensure that any gaps are covered. One of the hospitals serves a large South-East Asian population, some of whom do not speak or read English. To facilitate inclusion, a translator will work with researchers to approach and consent patients who speak Urdu and/or Potwari—the languages most commonly spoken among the largest non-English-speaking group in that area—and provide translation services during the course of the research.

We are excluding patients who are at the end of their life or whose care has become palliative, so as not to place additional burden on themselves or their families. We will, however, be approaching people with cognitive or language impairments, including patients who lack or have variable capacity to consent to the research for themselves, if they have suitable support in place to help them to participate in the research. This group of patients are likely to be especially vulnerable during the transition period; thus, it is particularly important to capture their experiences and those of the people who care for them to explore opportunities to reduce risk to this population. All the researchers working on the study have received additional training on taking informed consent in adults lacking capacity. When a patient is identified as not having the capacity to give consent, in line with the Mental Capacity Act 2005,[25] the researcher will take reasonable steps to identify a personal consultee to advise on the presumed wishes and feelings of participants unable to consent for themselves and on their inclusion and participation in the research. We will also seek to recruit the consultee as a participant in the study so that they can provide support to the patient-participant throughout the research process.

### Data collection
As part of a focused ethnographic approach,[26] we will employ the following methods to explore experiences and identify likely influences on outcomes:
► Non-participant observation, with discussions about 'key moments'.
► 'Go-Along' interviews.[27 28]
► Individual semistructured interviews.

These data collection methods will be combined flexibly within this study to enable us to gather rich insightful data into what patients think, feel and believe about the process of leaving hospital to return home. Two researchers will be responsible for data collection, each following the patients they recruit for their entire 'transitions journey' (where possible).

### Observations
Observations will be used to explore what happens to a patient at various time points and locations as they

transition from hospital to home, including within the admitting hospital, a transitional care facility, the patient's residence and other care settings. Non-participant observation offers a direct view of behaviours in their natural setting.[29 30] It allows the researcher insight into what is done, and how it is done, by various people involved in delivering care over the transition period (eg, healthcare professionals, support and administrative staff, the voluntary sector and patients and their carers themselves). Observations will provide the foundation for short informal conversations (approximately 10–15 min) to follow up on 'key moments' observed on a previous occasion. These will happen as close to the original event as possible to enable accurate recall. Observations and conversations will be captured through field notes. An observation framework will be developed for this study as a prompt for observer field notes, ensuring accurate, in-depth recording of observations and facilitating analysis.

### 'Go-Along' interviews

'Go-Along' interviewing is a participatory method that is person-centred and interactive, that is, they focus on understanding the experiences of a person within changing contexts in real time. Interviewing someone while they are experiencing something in real time can facilitate articulation of attachments, feelings and memories that might otherwise remain unconscious or unsaid.[27 28] With this in mind, the researcher will accompany the participant within the context in which care is being delivered, with all conversation recorded digitally. Recordings will be supplemented by field notes to provide context and aid interpretation of transcribed data.[28] We are aware that a 'Go-Along' interview may not be appropriate in all circumstances and so we will use this method sensitively according to the context in which the researcher and patient are in and what is happening at that time. For example, we will not observe intimate patient care such as using the toilet or showering. We will always be guided by what the participant (and those also present) is comfortable with and consents to.

### Interviews

Observations and 'Go-Along' interviews will be supplemented by more formal semistructured interviews that will use a guide (see online supplementary appendix 1) to provide a framework to the discussions. Informed by the 'capability', 'opportunity', 'motivation' and 'behaviour' (COM-B) framework,[31] this guide will contain some key questions addressing issues of capability, motivation and opportunity for patients to be involved in their care at transitions; it will also be informed by the observations that have occurred up to that point. The COM-B framework is particularly valuable as a tool for understanding the factors that act as both barriers and facilitators for behaviour prior to intervention development. If, for example, we were to identify that patients and their carers were rarely involved in their care, it is valuable, in terms

of targeting the intervention to understand whether this is because patients are unwilling to be involved (low motivation), they just do not feel they have the knowledge or skills (low capability) or that the formal carers dismiss attempts by patients to be involved (low opportunity). The COM-B complements our broader conceptualisation of transitions within a resilience framework because it focuses on understanding what patients actually do (work as done), rather than assuming that they do what is imagined (eg, by those caring for them). Interviews will be cogenerated by both participant and researcher; to ensure that discussions are relevant to the research, the researcher will use the interview schedule as a 'map' to guide the conversation, while remaining flexible enough to follow participants as they express their experiences about being in hospital and transitioning from hospital to home. Interviews will be recorded digitally. Individual interviews are likely to take place in the hospital and in the patient's own home; if an interview does take place in a setting that is not the patient's home, we will ensure that these occur in a space that is sufficiently private. We may also conduct telephone interviews to speak with participants about an episode of care that has been delivered but not observed by the researchers (eg, visiting their GP).

We expect that each of these methods will be used to gather data from each participant, but to remain sensitive to the needs of the patient or carer, the context within which healthcare is delivered and the needs of the research, we will employ them flexibly and sensitively. For example, sometimes, it may not be appropriate to use a more participatory approach, such as a 'Go-Along' interview, because it is important that we capture interactions between healthcare professionals and patients as they would naturally occur, without the participation of the researcher. Also, important care may be delivered and the participation of the researcher in the interaction would disrupt the delivery of that care (eg, within a rehabilitation therapy session). At other times, however, it may be helpful to use the time spent with patients as they are moving from one location to another, for example, capturing their thoughts, feelings and beliefs about what has been, and is, happening to them in that moment, alongside their expectations about what will happen in the future. Within this context, a more structured non-participant observation would likely fail to capture the richness of the patient's experience. More formal semistructured interviews will complement both types of observational work.

### Timing of data collection

'Time' and 'place' are two important features of any transition process. We have therefore designed the research to capture as much of the temporospatial aspects of the transition from hospital to home as possible. This includes collecting data from participants at various time points within the transitions process, and in various locations. It also involves exploring the significance of 'time' and 'place' with participants.

Data collection will be organised around five 'episodes', over a period of 3–4 months:

1. On, or shortly after, admission to hospital.
2. Shortly prior to and/or during discharge from the admitting hospital.
3. A day or two after discharge in the home or intermediate care.
4. Several weeks after discharge.
5. Three months after discharge or on readmission if sooner.

Data collection may occur within the admitting hospital, an intermediate care facility and in the home of the participant. In addition, if the patient gives us permission, we will follow the patients to appointments that form part of their 'discharge care package' (eg, appointments with therapists or district nurses). We anticipate that we will see each patient approximately five times (once within each 'transition episode'). However, the actual number of times that we will see the participant will be guided by the needs and experiences of the patient. For example, someone experiencing fatigue as an outcome of stroke may require more visits of a short duration to avoid placing unnecessary burden on the participant. Alternatively, some patients may have multiple appointments at the point of discharge and be happy for us to accompany them to each of these appointments. Data collection will remain sufficiently flexible to meet the needs of the participants and the research. We anticipate that all data collection will be complete by March 2018.

### Data analysis

All interviews will be digitally recorded and transcribed verbatim. Relevant contextual details will be added to the interview transcripts from notes made by the researcher. Researchers will make field notes during observations. After an observation session, the researchers will use a digital recorder to describe what they observed and to digitally capture their own interpretation of the session; this will then be transcribed verbatim. Transcription will be done by an external agency and checked by the researcher who collects the data.

Data analysis will be inductive and flexible, using a framework approach[9] to identify themes and analytical categories. Framework analysis allows the researcher to move from raw data to wider explanatory accounts through a series of conceptual groupings and meanings assigned to the data.[32 33] The key stages of framework analysis are: familiarisation with data; identifying a thematic framework; indexing and sorting data; reviewing and refining the thematic framework, and then summarising and displaying the data through the construction of thematic matrices.[34] These matrices allow the data to be reduced and distilled, while staying close to the original text. The matrices also facilitate comparison within—and between—themes and cases (participants). Within-case comparison will be particularly helpful when exploring the temporal aspects of the transitions process, as it will allow exploration of changes in individual attitudes and

experience over time. Data analysis will be conducted by both researchers involved in data collection.

The thematic frameworks will be constructed by both researchers, using the interview guide as a tool for organising the data. Each researcher will label and sort their own data using the thematic framework but discussion about emergent findings will happen on a regular basis and will be used to refine the thematic framework. The comparison work to identify analytical categories and explanatory accounts will be done together and will also involve members of the project patient panel. Qualitative data analysis software (NVivo V.10 for Windows) will be used to help manage and organise the data into thematic matrices.

### PATIENT AND PUBLIC INVOLVEMENT

The Yorkshire Quality and Safety Research Group currently supports a patient and public panel of 25 people representing the local patient community. This group has been involved from the beginning of the Partners at Care Transitions (PACT) research study and will continue to provide input when necessary. In addition, we have recruited a panel of people who will work with the PACT research team over the course of the study. Panel members will meet regularly as a group to support the PACT study as a whole; panel members will also be working in pairs to support one of the first three work-packages, including this study of patient experience. We anticipate that the PACT patient panel will contribute to the analysis and interpretation of research findings and to the development of the intervention in light of these findings. Panel members will be supported by a research nurse with expertise in patient and public involvement in research.

### ETHICS

Prior to approaching any patient, the researcher will speak with a senior healthcare professional to find out which patients may be approached to take part in the research. This is to ensure that we do not approach people who are very unwell or at the end of their life. At first approach, the researcher will be accompanied by a member of the clinical team, who will make the first introduction. All potential participants will be provided with: verbal and written information about the study; the opportunity to ask questions; and time to consider whether they would like to participate. Informed consent will be gained from all participants (patients and carers) who can consent for themselves. All research documents, such as information sheets and consent forms, are written in plain English using large print, and laid out clearly to facilitate readability and understanding. Verbal consent scripts will be used with people who struggle with written language or who have a physical impairment that prevents them from signing a consent form.

We recognise that consent is an ongoing process. Therefore, at every research encounter, we will check whether participants still wish to take part prior to starting any data collection. As far as possible, the same researcher will do all follow-up work with the same patient to promote building of a relationship and to avoid confusion for the older person and/or their carer. Participants will be free to withdraw from the study at any time and can choose whether the data collected about them are included in the analysis.

All personal identifiable data will be kept securely in line with legal requirements and best practice recommendations to ensure confidentiality. Participants will be assigned pseudonyms so that they cannot be identified.

When healthcare staff are present during an observation, verbal consent will be sought from the staff member at that time. If they agree to observation and/or audio-recording, the observation will continue as planned. If they do not agree to be observed, the researcher will seek to understand what the staff member is and is not comfortable with and proceed accordingly. For example, a member of staff may agree for a researcher to be present but would not like any details about them or their actions recorded in any way. In this circumstance, and with the patient's permission, the researcher may stay and observe but will not record any information about the staff member. If the staff member declines all observations, then the researcher will not observe the interaction and will follow up with research participants after the interaction is over and the staff member is no longer present.

## Safeguarding

Consent will be obtained on the understanding that all interactions are confidential unless the researcher witnesses actions that cause them to be concerned for an individual's safety. Should a researcher believe that a research participant (or other person) is at risk of harm, through observation or disclosure during an interview, the researcher will encourage the person to raise this with a relevant professional, or offer to raise it on their behalf. Should consent not be given by the person, if the researcher feels that the person is at risk, then the researcher will disclose the issue/incident without consent but in the interest of the person's safety and well-being. Guidance will be sought from local clinical collaborators regarding appropriateness to escalate concerns. In emergency or urgent situations (eg, witnessing a person fall, or experience life-threatening symptoms such as severe breathing difficulties), the researcher will immediately contact the appropriate emergency services.

## DISSEMINATION

The findings of the study will contribute to the other WP within the programme of work. Particular contributions include using the data: to inform the development (and subsequent testing) of a patient-centred intervention that aims to improve the transitions experience and reduce hospital readmissions (WPs 4, 5 and 6); and to inform the development of a measure of the quality of transitions, which will be used as a secondary outcome measure within the PACT randomised control trial (WPs 3 and 6).

We will also develop 'patient experience of transitions' resources in the form of anonymised stories to help communicate the main findings of the project to both academic and clinical groups. For example, the Academic Health Science Network Improvement Academy and educational institutions will be used to disseminate these resources to people undergoing training and/or quality improvement work. We will also be hosting a national conference to showcase findings from this project and two of the other linked WPs.

We will publish our research findings in academic and professional journals and present our work at relevant national and international conferences. We also plan to support dissemination through a website, social media and networks. We have experience of using these formats for reaching a variety of audiences, but particularly our local clinical networks. Twitter has proved a particularly effective method for sharing our ideas, alerting people to our recent findings and discussing new ideas and concepts.

Our dissemination strategy has been developed in partnership with various stakeholders, including our patient panel. We will continue to engage with and involve these groups to ensure that the research findings can be translated effectively into clinical practice and to maximise the impact of the research locally and nationally.

## DISCUSSION
### Strengths and weaknesses

This study seeks to explore and describe the experience of older people and their families as they transition from hospital to home. Using multiple in-depth qualitative research methods enables us to capture detailed accounts of experiences and perceptions of experiences, alongside the context within which care is occurring. Nonetheless, we recognise that observational methods have the potential to introduce bias into the study, because people (in this case, health service staff) may change their behaviour when they know they are being observed. However, in agreement with McNaughton Nicholls *et al*,[35] we believe that the strengths of observational methods, for example, access to rich data that would not be accessible otherwise, alongside insight into 'interactions, processes and behaviours that goes beyond… verbal accounts', outweigh the potential risk inherent within the research process.

The study design means that the findings will not be generalisable to all older people transitioning from hospital to home. Nonetheless, the research accounts have the capacity to provide data which are credible, dependable and transferable to others.[36] Moreover, Rossman and Rallis[37] argue that 'the ultimate goal of qualitative research is learning, that is, the transformation of data in to information that can be used. *Use can be considered*

*an ethical mandate.* The use of the findings of this study as a basis for a new patient-centred intervention can be considered to fulfil this ethical mandate and is thus a strength of this research.

The findings of the research will contribute to the development and testing of a person-centred intervention that aims to improve patient experience and reduce the risk of hospital readmission. It is anticipated that improving the patient experience of the transition process will contribute to improved safety and quality of care[11 38] during this transition period. It is also anticipated that providing good transitional care will reduce hospital readmissions. This has benefits for patients and their families, as being in hospital is associated with a number of risks and has a psychological and physical impact on patients and their families.[13 15] Risks such as hospital-acquired infections are increased, for example, and issues such as disrupted sleep, nutritional deficiencies and problems caused by poor nourishment, increased stress and anxiety and deconditioning due to inactivity and bed rest can place additional burdens on people already dealing with one or more conditions or trauma.[7] Reducing readmissions also has benefits for the health service which is under pressure to deliver more care with less resource. Moreover, NHS Trusts now incur financial penalties for readmissions within 30 days; reducing readmissions would reduce spending on such penalties.

We want to learn from older people and their families about what works for them in the care that they receive and to find out what would improve their experience of the transitions process. Exploring the transition process from their perspective, particularly looking at where and how people can be involved in their care, and using these data to develop an intervention, means that the patient is at the heart of quality improvement. This research will also add to an existing body of knowledge about patient experiences of care at transitions.[14 16–20] Importantly, this research will capture the temporospatial experiences of transitions by following older people and their families during their transition journey from admission through to 3 months postdischarge. This element is missing from existing research, most of which captures patient experience data at only one time point. Moreover, much of the existing research exploring patient experience data about care at transitions appears to capture what goes wrong, or the ways in which individuals are dissatisfied with the care they receive. Conversely, our research will be exploring what goes well at transitions of care, as well as seeking to identify areas for improvement. By doing so, we will add an important dimension to the growing knowledge base about care at the transition from hospital to home. Also, the adoption of a resilience-engineering approach to safety acknowledges the positive contribution that all people can make to the delivery of good quality, safe healthcare, and engenders the harnessing of a genuine partnership to improve patient experience and clinical outcomes.

**Author affiliations**
[1]Yorkshire Quality and Safety Research Group, Bradford Institute for Health Research, Bradford, UK
[2]Social Policy Research Unit, University of York, York, UK
[3]Leeds Centre for Older People's Medicine, Leeds Teaching Hospitals NHS Trust, Leeds, UK
[4]School of Psychology, University of Leeds, Leeds, UK

**Acknowledgements** The research is supported by the NIHR CLAHRC Yorkshire and Humber.

**Contributors** RL, AC, LS and YB designed the overall programme of research and conception of studies within. NH, YB, JM, LS, LH, JH, AC and RL were involved in the design of the current study and have contributed to the drafting, reviewing and final approval of the manuscript.

**Funding** This report is an independent research funded by the National Institute for Health Research (National Institute for Health Research Programme Grants for Applied Health Research, Partners at Care Transitions (PACT): Improving patient experience and safety at transitions in care, RP-PG-1214-20017.

**Disclaimer** The views expressed in this publication are those of the author(s) and not necessarily those of the NHS, the National Institute for Health Research or the Department of Health.

**Competing interests** None declared.

**Ethics approval** This study has been approved by the Wales 7 Research Ethics Committee (reference: 17/WA/0057).

**Provenance and peer review** Not commissioned; externally peer reviewed.

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
