## [Reviewer comments · BMJ Open]

ARTICLE DETAILS

TITLE (PROVISIONAL)	Partners At Care Transitions (PACT). Exploring older peoples' experiences of transitioning from hospital to home in the UK: protocol for an observation and interview study of older people and their families to understand patient experience and involvement in care at transitions.
AUTHORS	Hardicre, Natasha; Birks, Yvonne; Murray, Jenni; Sheard, Laura; Hughes, Lesley; Heyhoe, Jane; Cracknell, Alison; Lawton, Rebecca

VERSION 1 – REVIEW

REVIEWER	Gavin Daker-White The University of Manchester, UK
REVIEW RETURNED	21-Jun-2017

GENERAL COMMENTS	This is a well described protocol using established methods. I have some minor concerns around referencing, ethics and the proposed sample size. Referencing The first paragraph of the "Data collection" section (p.15) appears to lack references. I realise that this paragraph is introductory, but in my opinion, phrases such as "focused ethnographic approach," " 'Go-Along' interviews" and "transitions journey" might usefully be referenced here. I wasn't familiar with the expression "Go-Along interviews" but the method as described appears akin to "shadowing" which is the term more usually employed in ethnographic research. How are "Go-Along interviews" distinct from shadowing ? A minor point for discussion perhaps. This is an active research field and I wondered whether the reference list needs expanding and updating to included more recent work relevant to the research questions being posed. I am aware of at least 3 relevant published articles not included, although there will be more: 1. Rustad, Else Cathrine, Bodil Furnes, Berit Seiger Cronfalk, and Elin Dysvik. "Older patients' experiences during care transition." Patient preference and adherence 10 (2016): 769.2. Gregersen, Merete, Anita Haahr, Lene Holst Pedersen, and Else Marie Damsgaard. "Patient satisfaction and early geriatric follow-up after discharge in older acute medical patients." Clinical Nursing Studies 4, no. 3 (2016): p78.
--

3. Bångsbo, Angela, Eva Lidén, and Anna Dunér. "Patient participation in discharge planning conference." *International journal of integrated care* 14, no. 4 (2014).

Ethics

The research team will be observing patients in interaction with healthcare workers. Whilst there is exhaustive description of consent procedures in relation to participating patients, there is no discussion of how or whether staff consent will be sought to observe the same situations. What if staff are not comfortable being observed or having a researcher present? Are there any circumstances in which it would not be appropriate for the researchers to observe certain interactions or procedures? In what circumstances, if any, will they have to withdraw?

Proposed sample size

30 patients will be recruited from six different departments across 2 hospitals. The researchers wish to identify different "transitional challenges" and diversity in terms of gender, ethnicity, age, type of admission, acute vs. chronic illness, polypharmacy, multimorbidity, language ability and whether informal carers are present or not. Will the number of participants be sufficient to explore this large range of different factors? The use of opportunistic (they use "opportunity") followed by purposive sampling implies a grounded theory approach to recruitment. But there is no mention of data saturation. In short, I am unclear about the rationale for seeking to recruit "30" participants and no reference is provided to back up this figure.

Other minor issues

1. The stated appendix (p.16, line 14) was not present and I was thus unable to comment on the interview topic guide.

2. I did wonder about "contamination" in healthcare procedures given the presence of researchers. Whilst this is not a fundamental flaw, perhaps it requires a little discussion?

3. Given the stated research aims, I wondered whether an action research approach would have been appropriate? More importantly, what is the protocol for the research team should they encounter serious harms or adverse events (e.g. post discharge) which they feel require either urgent intervention or notification to the participant's General Practitioner?

4. The team appear to view care transitions as conceptually involving a singular move from hospital to home. However, we know from the literature and current media reports that some patients may find themselves in a "revolving door" of repeated admission and discharge.

REVIEWER	Dr Anita Slade Institute of Applied Health Research College of Medical and Dental Sciences University of Birmingham UK
REVIEW RETURNED	24-Jul-2017

GENERAL COMMENTS	Overall this is a well written paper that clearly discusses your proposal. However, I feel the abstract is not well written in comparison to the rest of the paper and needs some attention. I suspect you have tried to keep within the word limit and this has resulted in something that appears disjointed and doesn't make sense in place e.g. line 12 of abstract. Please review the abstract. Something which I also feel needs to be clarified is whether you are going to let staff know that you are observing them for a research project or whether they will be informed that the patient is part of a research project as I feel this is both a potential ethical issue but could also produce a potential bias and change in behaviour of the health care professionals if they knew that you were reviewing 'good care'. You have not addressed this at all in your methodology, ethics or discussion and I feel this needs to be addressed before the paper can be published. I feel unable to comment on the 'Go-along' interview method as I am not familiar with this form on interviewing so I feel the paper should be reviewed by someone who is familiar with this methodology. I also feel you need to discuss potential issues of using a translator as this might also introduce bias and loss of important cultural references that might not translate. I would suggest it might be important for a qualitative research who is also a native speaker to listen to transcripts and identify potential themes. This is only a suggestion and not a condition of the review but I know from experience that identifying themes from the original language identifies 'themes' that might be lost if using a translated version from a third party.
--

REVIEWER	Lianne Jeffs St. Michaels Hospital Canada
REVIEW RETURNED	28-Jul-2017

GENERAL COMMENTS	Overall a well written protocol paper, however it is not clear what new insight will be generated as there is growing body of knowledge around patient and families views on care transitions (you mention a review) moving to co-design of interventions (e.g. PODS in Canada). There is a brief mention of resilience engineering but not details on the key concepts anyhow they frame the inquiry - is this a novel framing? resilience engineering and safety and quality in health care is not new. Is your methodology novel - it combines qualitative methods over time - is there something novel that might emerge from this type of design? For your observation do you have a matrix from resilience engineering and the body of literature to assist in sorting and coding data and field notes?
--

VERSION 1 – AUTHOR RESPONSE

Reviewer 1

Comment: The first paragraph of the "Data collection" section (p.15) appears to lack references. I realise that this paragraph is introductory, but in my opinion, phrases such as "focused ethnographic approach," " 'Go-Along' interviews" and "transitions journey" might usefully be referenced here.

Response: 'Focused ethnographic approach' reference added; 'Go-along interviews' references added; 'Transitions journey' is a term we are using to describe the experience of the transition phase over time, therefore there is no appropriate reference to cite here.

Comment: I wasn't familiar with the expression "Go-Along interviews" but the method as described appears akin to "shadowing" which is the term more usually employed in ethnographic research. How are "Go-Along interviews" distinct from shadowing ? A minor point for discussion perhaps.

Response: We recognise that shadowing as a form of observation is very similar to the Go Along interview method. We see Go-Along interviews as a more purposive and focused form of shadowing, in which there may be more interaction between the participant and the researcher than with typical shadowing.

Comment: This is an active research field and I wondered whether the reference list needs expanding and updating to included more recent work relevant to the research questions being posed. I am aware of at least 3 relevant published articles not included, although there well be more:

Response: Thank you for this comment. We have updated the reference list with three additional more recent and very relevant references (Rustad et al, 2016; Neiterman et al, 2015; Hvalvik and Dale 2015).

Comment: The research team will be observing patients in interaction with healthcare workers. Whilst there is exhaustive description of consent procedures in relation to participating patients, there is no discussion of how or whether staff consent will be sought to observe the same situations. What if staff are not comfortable being observed or having a researcher present?

Response: We have added a paragraph on page 10 & 11 about seeking verbal consent from staff: When healthcare staff are present during an observation, verbal consent will be sought from the staff member at that time. If they agree to observation and/or audio-recording, the observation will continue as planned. If they do not agree to be observed, the researcher will seek to understand what the staff member is and is not comfortable with and proceed accordingly. For example, a member of staff may agree for a researcher to be present but would not like any details about them or their actions recorded in any way. In this circumstance, and with the patient's permission, the researcher may stay and observe but will not record any information about the staff member. If the staff member declines all observation, then the researcher will not observe the interaction and will follow up with research participants after the interaction is over and the staff member is no longer present.

Comment: Are there any circumstances in which it would not be appropriate for the researchers to observe certain interactions or procedures? In what circumstances, if any, will they have to withdraw?

Response: We have added the following sentence on page 7 'For example, we will not observe intimate patient care such as using the toilet or showering. We will always be guided by what the participant (and those also present) are comfortable with and consent to.'

Comment: 30 patients will be recruited from six different departments across 2 hospitals. The researchers wish to identify different "transitional challenges" and diversity in terms of gender, ethnicity, age, type of admission, acute vs. chronic illness, polypharmacy, multimorbidity, language ability and whether informal carers are present or not. Will the number of participants be sufficient to explore this large range of different factors? The use of opportunistic (they use "opportunity") followed by purposive sampling implies a grounded theory approach to recruitment. But there is no mention of data saturation. In short, I am unclear about the rationale for seeking to recruit "30" participants and no reference is provided to back up this figure.

Response: We have added the following on page 5:

"We anticipate that a sample of 30 patients is likely to allow us to capture some diversity and is also likely to achieve theoretical saturation; however, this will be reviewed as analysis proceeds to ensure any gaps are covered."

Comment: The stated appendix (p.16, line 14) was not present and I was thus unable to comment on the interview topic guide.

Response: Apologies, this was an oversight. We have now included it as a supplementary file.

Comment: I did wonder about "contamination" in healthcare procedures given the presence of researchers. Whilst this is not a fundamental flaw, perhaps it requires a little discussion?

Response: We have added a point to the 'strengths and weaknesses' sections (after the abstract on page 3 and on page 12).

Page 3: Although non-participant observation can generate rich contextual data that are not as easily accessed via other methods, the presence of a researcher has the potential to affect the behaviour of those being observed.

Page 12: Nonetheless, we recognise that observational methods have the potential to introduce bias into the study, because people (in this case, health service staff) may change their behaviour when they know they are being observed. However, in agreement with McNaughton Nicholls et al, 2014, [34] we believe that the strengths of observational methods, e.g. access to rich data that would not be accessible otherwise, alongside insight into "interactions, processes and behaviours that goes beyond... verbal accounts", outweighs the potential risk inherent within the research process.

Comment: Given the stated research aims, I wondered whether an action research approach would have been appropriate?

Response: Thank you for your comment regarding the potential utility of an action research approach. It is an approach we are currently using within another of our active projects at the moment and is generating interesting work. In the PACT project, however, we have decided to focus on collecting data from multiple sources (e.g. patients and carers, health care professionals, and research literature) and gathering them together to develop an intervention that synthesises these multiple viewpoints. Consequently, we have opted for separate data collection projects (work package 1, work package 2, and review work) and then drawing them together in work package 4 (the intervention development work package).

Comment: More importantly, what is the protocol for the research team should they encounter serious harms or adverse events (e.g. post discharge) which they feel require either urgent intervention or notification to the participant's General Practitioner?

Response: We have added a paragraph on page 11 under a new 'safeguarding' heading:
"Consent will be obtained on the understanding that all interactions are confidential unless the researcher witnesses actions which cause them to be concerned for an individual's safety. Should a researcher believe that a research participant (or other person) is at risk of harm, through observation or disclosure during an interview, the researcher will encourage the person to raise this with a relevant professional, or offer to raise it on their behalf. Should consent not be given by the person, if the researcher feels that the person is at risk then the researcher will disclose the issue/incident without consent but in the interest of the person's safety and well-being. Guidance will be sought from local clinical collaborators regarding appropriateness to escalate concerns. In emergency or urgent situations (e.g. witnessing a person fall, or experience life-threatening symptoms such as severe breathing difficulties), the researcher will immediately contact the appropriate emergency services."

Comment: The team appear to view care transitions as conceptually involving a singular move from hospital to home. However, we know from the literature and current media reports that some patients may find themselves in a "revolving door" of repeated admission and discharge.

Response: Thank you for this comment – we have now clarified our view of transitions by adding a couple of sentences on page 3.

"Additionally, older people may experience more than one 'transition' in a single hospital admission episode, for example, moving between wards or via intermediate care at a different location. Likewise, some older people may experience readmissions within a short period of time. The transitions process may not, therefore, be a linear one, and resulting in further complexity."

Reviewer 2

Comment: Overall this is a well written paper that clearly discusses your proposal. However, I feel the abstract is not well written in comparison to the rest of the paper and needs some attention. I suspect you have tried to keep within the word limit and this has resulted in something that appears disjointed and doesn't make sense in place e.g. line 12 of abstract. Please review the abstract.

Response: We have amended the abstract as follows:

"Introduction: Lengths of hospital inpatient stays have reduced. This benefits patients, who prefer to be at home, and hospitals, which can treat more people when stays are shorter. Patients may, however, leave hospital sicker, with ongoing care needs. The transition period from hospital to home, can be risky, particularly for older patients with complex health and social needs. Improving patient experience, especially through greater patient involvement, may improve outcomes for patients and is a key indicator of care quality and safety. In this research we aim to: capture the experiences of older patients and their families during the transition from hospital to home; and identify opportunities for greater patient involvement in care, particularly where this contributes to greater individual- and organisational-level resilience.

Methods and Analysis: A 'focused ethnography' comprising observations, 'Go-Along' and semi-structured interviews will be used to capture patient and carer experiences during different points in the care transition from admission to 90 days after discharge. We will recruit 30 patients and their carers from six hospital departments across two NHS Trusts. Analysis of observations and interviews will use a Framework approach to identify themes to understand the experience of transitions and generate ideas about how patients could be more actively involved in their care. This will include exploring what 'good' care at transitions look like and seeking out examples of success, as well as recommendations for improvement.

Ethics and dissemination: Ethical approval was received from the NHS Research Ethics Committee in Wales. The research findings will add to a growing body of knowledge about patient experience of transitions, in particular providing insight into the experiences of patients and carers throughout the transitions process, in 'real time'. Importantly, the data will be used to inform the development of a patient-centred intervention to improve the quality and safety of transitions.”

Comment: Something which I also feel needs to be clarified is whether you are going to let staff know that you are observing them for a research project or whether they will be informed that the patient is part of a research project as I feel this is both a potential ethical issue but could also produce a potential bias and change in behaviour of the health care professionals if they knew that you were reviewing 'good care'. You have not addressed this at all in your methodology, ethics or discussion and I feel this needs to be addressed before the paper can be published.

Response: Thank you for your comment. We have added a section on page 11 stating that we are seeking verbal consent to observe staff when patient care is being delivered. We have also added a comment within the strengths and limitations sections (page 3 and page 12) acknowledging the potential biases that exist when observing/researching 'usual care'.

Comment: I also feel you need to discuss potential issues of using a translator as this might also introduce bias and loss of important cultural references that might not translate. I would suggest it might be important for a qualitative research who is also a native speaker to listen to transcripts and identify potential themes. This is only a suggestion and not a condition of the review but I know from experience that identifying themes from the original language identifies 'themes' that might be lost if using a translated version from a third party.

Response: Thank you for highlighting some important points regarding using a translator within this work. The translator we have recruited for this research project is a native speaker of the languages that we are asking her to work in. She will be completing the transcription of those interviews conducted in Urdu or Potwari in order to capture as much of the original meaning as possible. She will also be present during the data analysis of these transcripts discuss to guide our interpretation and identification of themes.

Reviewer 3

Comment: Overall a well written protocol paper, however it is not clear what new insight will be generated as there is growing body of knowledge around patient and families views on care transitions (you mention a review) moving to co-design of interventions (e.g. PODS in Canada). There is a brief mention of resilience engineering but not details on the key concepts anyhow they frame the inquiry - is this a novel framing? resilience engineering and safety and quality in health care is not new. Is your methodology novel - it combines qualitative methods over time - is there something novel that might emerge from this type of design? For your observation do you have a matrix from resilience engineering and the body of literature to assist in sorting and coding data and field notes?

Response: Thank you for your comments regarding the lack of clarity about what new insights will be generated through the research. We have added a few sentences on pages 12 and 13.

Page 12: “The use of the findings of this study as a basis for a new patient-centred intervention can be considered to fulfil this ethical mandate and is thus a strength of this research.”

Page 13: “This element is missing from existing research, most of which captures patient experience data at only one time point. Moreover, much of the existing research exploring patient experience data about care at transitions appears to capture what goes wrong, or the ways in which individuals are dissatisfied with the care they receive.

Conversely, our research will be exploring what goes well at transitions of care, as well as seeking to identify areas for improvement. By doing so, we will add an important dimension to the growing knowledge base about care at the transition from hospital to home.”

We have also slightly expanded our discussion of resilience to provide more clarity about our thinking around this concept. There is a recognition within discussions of resilience and resilience engineering that the concept is complex and has not yet been fully operationalized (e.g. Woods et al, 2015 & Anderson et al, 2015). With this in mind, we are currently in discussion about how best to apply this synergistically and consistently across our work packages within our programme of work. For these reasons and given that this is a protocol paper we wish to inform the readers of our broad intentions, leaving the detail for our findings paper. Nonetheless, we appreciate that we have not provided enough detail to make our thinking clear – we hope the following adds a little more clarity.

"The programme of work utilises a resilience engineering approach to safety in healthcare,[18] and we especially want to learn from what goes well at transitions, rather than focusing only on what goes wrong; doing so “sheds light on otherwise unrecognised and unspecified pathways to success”. [19] Within this project, we want to understand the things that patients, relatives and health service staff (or others) do to enable patients and their families to be resilient within the transitions process. However, we also want to explore the ways in which patients and their carers do or could contribute to organisational resilience. Schubert et al, for example, suggest that patients/caregivers can “identify and prevent mistakes from happening, and participate in improving their care” by navigating a “fragmented system” through the co-ordination of tasks across multiple health care settings and providers. This will enable us to take a proactive approach towards care during the transitions period; developing an intervention that helps to inform people about what they can do to make the transitions process ‘good’. We believe this is a novel approach towards understanding and improving care at the transitions period." (Page 4-5)

Our data is going to be coded inductively, as mentioned on page 9 of the paper, so we do not have a matrix from RE that we are coding data to, but thank you for your suggestion – it is something we may explore if necessary.

VERSION 2 – REVIEW

REVIEWER	Gavin Daker-White University of Manchester, UK
REVIEW RETURNED	11-Sep-2017

GENERAL COMMENTS	I am pleased to see that the authors have responded to the concerns raised following the first version and am now happy to recommend that the manuscript be accepted for publication.
---

REVIEWER	Anita Slade Centre for Patient Reported outcomes University of Birmingham
REVIEW RETURNED	19-Sep-2017

GENERAL COMMENTS	Overall the authors have addressed my previous concerns, although I think the discussions on limitations could have gone into more depth on the implications of observing someone and how it will change behaviours. There are a small number of minor edits: Line 54 page 2 doesn't make sense and needs reworking Page 3 Lines 19 to 25 is all one sentence this is too long and needs breaking up. Ditto page 7 lines 43 to 48 Otherwise I feel this is an acceptable piece of work that covers an important topic.
--

REVIEWER	Lianne Jeffs St Michaels Hospital, Canada
REVIEW RETURNED	28-Sep-2017
GENERAL COMMENTS	The reviewer also provided a marked copy with additional comments. Please contact the publisher for full details.

VERSION 2 – AUTHOR RESPONSE

Reviewer 2

Reviewer comment:

There are a small number of minor edits:

Line 54 page 2 doesn't make sense and needs reworking

Page 3 Lines 19 to 25 is all one sentence this is too long and needs breaking up.

Ditto page 7 lines 43 to 48

Response:

Line 54 on page 2 has been amended: "The longitudinal approach enables us to gain insight into how patient experience and involvement change over time."

Lines 19-25 on page 3 have been amended: "Shorter stays in hospital have benefits for both patients, who prefer to be at home, and hospitals, which can treat more patients if stays are shorter. However, reduced stays can also result in an increased reliance on care outside the inpatient setting, for example, wound or catheter care, changes to medication, or input from therapy services."

Lines 43-48 on page 7 have been amended: "Informed by the COM-B framework[30], this guide will contain some key questions addressing issues of capability, motivation and opportunity for patients to be involved in their care at transitions; it will also be informed by the observations that have occurred up to that point."

Reviewer 3

Reviewer comment:

Not clear what gap exists and what your study will add?

Response:

We have added a paragraph to page 4 to explain how we think our study contributes new data and findings to the study of care transitions: "There are several published studies that have explored patient and carer perspectives on care at transitions [13-20]. However, much of this work appears to capture people's experiences at a single time point, often retrospectively after discharge. However, this study will recruit people whilst in the inpatient hospital setting, and follow them until approximately three months post-discharge. The longitudinal nature of the study will enable us to capture continuity and change in experience and involvement over time and will thus contribute new data and findings to a growing body of literature on care at transitions."

Reviewer 3 also made some minor editing suggestions regarding the addition or omission of words.

Response: We have made some of the changes as suggested. However, we have not altered the abstract because it is at the word limit and additional words were suggested. Although we agree that the additional words would have made the text more readable, we do not feel that the meaning would have been changed by these additions. Therefore, we do not feel that the omission of the suggested words will impact on the reader's understanding of the text.

Reviewer 3 also suggested that we remove the words 'thereby prolonging the overall patient stay' on page 3, line 53. We have decided to keep this because we believe that it is a salient point. We have, however, slightly changed the wording of the subsequent sentence to avoid repetition of the words 'patient stay'.

VERSION 3 – REVIEW

REVIEWER	Lianne Jeffs St Michaels Hospital Canada
REVIEW RETURNED	18-Oct-2017
GENERAL COMMENTS	Overall the paper has improved - what is not clear is how resilience framed your study/is framing your study and more description is required and linked to why COM-B which is a behavioural approach is also being used. Re-word generalizability to transferability and sampling needs more detail - what is opportunity? reads more that it is purposeful sampling using maximum variation and then delineate the characteristics you plan to cover to ensure a heterogenous sample.

VERSION 3 – AUTHOR RESPONSE

In light of the additional comments by Reviewer 3, we have made the following changes. We hope these are consistent with the recommendations made.

1. We have added these sentences to clarify how we think resilience is framing our study. "Moreover, the programme of work utilises a resilience engineering approach to safety in healthcare.[21] We especially want to learn from what goes well at transitions, rather than focusing only on what goes wrong; doing so “sheds light on otherwise unrecognised and unspecified pathways to success”. [22] Within this project, we want to understand resilience at two levels: 1) how patients and carers themselves bounce back, adapt and essentially cope with the transition process and what helps them to do this; and 2) how do patients and relatives get involved to prop up the transition process, in other words what work do they, and their informal and formal carers do to adapt to and overcome obstacles arising from a less than ideal system (e.g. discharge letters arriving at primary care days after discharge). In this latter case we will explore the ways that those people involved in the transitions process contribute to system resilience." (pages 4-5)

2. We have changed the description of the sampling, as follows:

"Sampling aims to capture maximum variation in respondents. We will purposively aim to recruit a diverse group of patients from different ethnicities, and gender groups, as well as a variety of ages – including the 'oldest old' (aged 85+) – wherever possible. We will also try to ensure that people with and without carers are included in the research, as carer involvement is likely to have an impact on the patient's experience of transition. Although sampling will be purposive, we recognise that in this context and population there is likely to be a degree of opportunistic recruitment; initially, the researchers will speak to clinical staff on each ward to identify eligible patients, selecting those who meet the criteria and who are available to approach at that time. The diversity of the sample will be monitored as participants are recruited. We anticipate that a sample of 30 patients is likely to allow us to capture some diversity and is also likely to achieve theoretical saturation; however, this will be reviewed as analysis proceeds to ensure any gaps are covered. One of the hospitals serves a large South-East Asian population, some of whom do not speak or read English. To facilitate inclusion, a translator will work with researchers to approach and consent patients who speak Urdu and/or Potwari – the languages most commonly spoken amongst the largest non-English speaking group in that area – and provide translation services during the course of the research." (page 6).

3. We have added the following sentences about the COM-B framework and how it fits in with the resilience approach we are using.

"The COM-B framework is particularly valuable as a tool for understanding the factors that act as both barriers and facilitators for behaviour prior to intervention development. If, for example, we were to identify that patients and their carers were rarely involved in their care, it is valuable, in terms of targeting the intervention to understand whether this is because patients are unwilling to be involved (low motivation), they just don't feel they have the knowledge or skills (low capability) or that the formal carers dismiss attempts by patients to be involved (low opportunity). The COM-B complements our broader conceptualisation of transitions within a resilience framework because it focuses on understanding what patients actually do (work as done), rather than assuming that they do what is imagined (by those caring for them, for example)." (Page 8).

4. We have not made any changes to the paragraph about generalizability/transferability on page 13. We would value some clarity from the reviewer about what she feels needs changing. We interpret her comment to suggest that we ought to use the word/concept of 'transferability' instead of 'generalisability' ("Re-word generalizability to transferability"). However, we already suggest in the paper that "The study design means that the findings will not be generalisable to all older people transitioning from hospital to home. Nonetheless, the research accounts have the capacity to provide data which are credible, dependable and transferable to others." (page 13). Apologies, if we have misinterpreted the comment - some additional detail would enable to adequately address the reviewer's concerns.

VERSION 4 – REVIEW

REVIEWER	Lianne Jeffs St Michaels Hospital Canada
REVIEW RETURNED	27-Oct-2017
GENERAL COMMENTS	The authors have addressed the comments from my last review - would recommend to add a reference to your description of maximum variation sampling.

VERSION 4 – AUTHOR RESPONSE

We have added the following reference to page 6. We hope this addresses the reviewers final recommendation.

24. Creswell JW. Qualitative Inquiry and Research Design: Choosing among Five Approaches. 3rd Ed. London: Sage, 2013.